behaviour/ecology/evolution

predator avoidance, deception, nest usurping, feathers, birds, hole-nesting

**Author for correspondence:**
Tore Slagsvold
e-mail: tore.slagsvold@ibv.uio.no

# Nest decoration: birds exploit a fear of feathers to guard their nest from usurpation

Tore Slagsvold[1] and Karen L. Wiebe[2]

[1]Centre for Ecological and Evolutionary Synthesis (CEES), Department of Biosciences, University of Oslo, Oslo 0316, Norway
[2]Department of Biology, University of Saskatchewan, Saskatoon, Canada

 TS, 0000-0003-2410-3269

Many species of birds incorporate feathers into their nest as structural support and to insulate the eggs or offspring. Here, we investigated the novel idea that birds reduce the risk of nest usurpation by decorating it with feathers to trigger a fear response in their rivals. We let prospecting birds choose between a dyad of nest-boxes in the wild, both containing some nest materials, but where one had a few white feathers and the other had none. All three species of cavity-nesting birds studied, the pied flycatcher *Ficedula hypoleuca*, the blue tit *Cyanistes caeruleus*, and the tree swallow *Tachycineta bicolor*, hesitated to enter boxes with white feathers. A similar avoidance of white feathers was found when the alternative nest-box of a dyad held black feathers. However, the birds readily collected white feathers that we placed in front of their nest-box, showing the fear of such feathers was context-dependent. We suggest that naive prospecting birds may perceive feathers in nests as the result of a predation event, and that owners decorate nests with bright feathers that can be seen from the opening to deter others from entering.

## 1. Introduction

Breeding birds may suffer heavily from predation, selecting for a number of behaviours to avoid the risk [1,2], including choice of habitat and nest site [3,4] and behaviour at the nest [5,6]. There may be strong competition for the safest sites among secondary cavity-nesting birds which try to escape predation by avoiding cavities with large entrances where the risk of nest predation is greater [7,8]. Competition for cavities may force birds to investigate unfamiliar, dark holes which may be dangerous to enter because an attacking animal may be hiding inside, such as stinging insects, poisonous herptiles, rodents, mustelids or raptors.

Mammals and avian predators, like small owls, may use cavities for plucking and eating avian prey [9], often leaving prey remains, faeces, urine and scents. Predators may also

(a)   (b)

**Figure 1.** Female blue tit incubating and decorating her nest with feathers. The presence of large, conspicuous feathers on the nest surface may be perceived by intruders as prey remains, causing prospecting birds to hesitate to enter even when no owner is present. (Photo: Roger Engvik.)

revisit cavities and kill both parents and offspring [10,11]. Birds may avoid cavities with remains from a visit of a mammal as shown by studies of birds where the content of the cavities used for roosting in winter and for nesting in spring was manipulated [12,13]. Tests of whether birds avoid only chemical cues in cavities have shown mixed results [13–16] compared with those presenting both visual and chemical cues of predators such as hair and faeces from mustelids [12,17].

Here we studied whether feathers in cavities may deter prospecting birds from a nest site. Birds may incorporate feathers into their nests to support the contents and to reduce heat loss but some large and conspicuous feathers which appear not only in the nest cup, but on the surface seem decorative (figure 1), suggesting a function in addition to insulation (an extended phenotype [18–21]). We propose a new idea, that birds decorate nests with feathers to trigger a fear response in their rivals, thereby reducing the risk of nest usurpation (below termed the Fear of Feathers Hypothesis).

Previous studies have shown that light intensity drops quickly with nest cavity depth and that colour vision is impaired in deep cavities [22]. Low light probably explains the light-coloured gapes of nestlings in cavities which help the parents to locate offspring in dim light [23,24]. Therefore, we assume that prospecting birds have difficulty seeing the bottom of cavities from the entrance hole and judging the contents without entering, explaining the hesitation to enter unfamiliar, dark-painted nest-boxes [25]. Thus, our hypothesis explains why the feathers used for nest decoration are often light-coloured and conspicuous [18,26,27] if they are meant to be readily seen from the entrance by the opponents.

We tested the Fear of Feathers Hypothesis in three species of secondary cavity-nesting passerine birds in the wild: the pied flycatcher *Ficedula hypoleuca*, the blue tit *Cyanistes caeruleus* and the tree swallow *Tachycineta bicolor*, representing three families with different ecologies and behaviours, to see whether avoidance of a nest decorated with conspicuous feathers is a general pattern. Pied flycatchers do not use feathers as lining material, whereas blue tits and tree swallows use many. Thus, pied flycatchers may be more reluctant to enter a cavity containing feathers in general. On the other hand, whereas blue tits are resident, pied flycatchers and tree swallows are migrants. Migrants which have less information about alternative nest sites than resident birds may be more willing to accept an unfamiliar nest cavity. Pied flycatchers even prefer a cavity that contains an abandoned nest versus an empty cavity, probably to save nest-building costs [28]. However, all three species are at risk of predation and hence should avoid entering unfamiliar cavities with feathers, in particular if the feathers are easily seen from the entrance. The feathers may not only reflect a recent killing of a bird but also obscure the view from the entrance hole, making it difficult to assess whether an aggressive nest owner, or a predator, is inside. Blue tits have patches of white on their forehead, cheeks and wings which blend in with the white feathers collected (figure 1).

To test the Fear of Feathers Hypothesis, we let birds choose between two newly erected and adjacent nest-boxes with some nest material but one with white feathers and one with no feathers. We also conducted dyad trials where the birds chose between nest-boxes with either white or black feathers to study whether it was the colour or the feathers themselves that caused an avoidance response. Similar trials were conducted with white feathers versus pieces of white paper to study whether the birds

feared white objects in general or only objects like feathers which are more likely indicators of prey remains. In the experiments, we used few feathers because only a few remains seem sufficient to simulate cues from mammal visits [29]. However, we also let pied flycatchers and blue tits choose between two nest-boxes where one held three and the other held six downy feathers of wood pigeons *Columba palumbus* because it may be easier to discover feathers from the entrance if there are many. In most experiments, we used commercial feathers to standardize their colour and form between trials. In a separate experiment, we offered these commercial feathers on the ground near blue tit nests to see whether the birds avoided them in general or only when seen inside a dark cavity.

## 2. Methods

### 2.1. Study area and study species

Pied flycatchers and blue tits were studied during 2019–2021 in mixed deciduous and coniferous woodlands near Oslo, Norway (approx. 59°56′ N, 10°32′ E) where nest-boxes have been provided for a number of years. The wooden nest-boxes had a 32 mm diameter entrance hole and a depth of 12–14 cm and were attached approximately 1.5 m above the ground to live trees. Pied flycatchers arrive on the study area after spring migration in late April to early May and lay their first eggs in the second half of May, whereas most blue tits are resident and peak egg-laying is at the end of April. Blue tits defend territories whereas pied flycatchers defend nest cavities but not feeding territories [30]. In both species, only the female builds the nest [30,31]. Pied flycatchers use mainly dry leaves, straw and thin bark from trees [30]. Blue tits use lining materials of hair, fur and feathers on top of a thick layer of moss [31].

The study of tree swallows was conducted during 2020–2021 in open grassland in British Columbia, Canada (approx. 51°38′ N, 121°18′ W). The wooden nest-boxes had a 40 mm diameter entrance hole and a depth of 18 cm, and were attached to fence posts. Tree swallows typically arrive in April and start egg-laying in May. They defend nest cavities but not feeding territories [32]. The base layer of tree swallow nests is mainly dry grass and the nest cup is lined with feathers. The female builds the basic structure and both sexes may gather feathers for the lining [33,34].

### 2.2. Dyad trials

Our experimental design has been used previously to study nest-box choice in pied flycatchers [25]. We selected unmated male pied flycatchers that had settled at a box in our study area and were singing to attract a female. In this species, the male almost stops singing after having attracted a female and she starts to build the nest very soon. The males could easily be separated from females by their dark dorsal colour [30]. For blue tits and tree swallows, we used pairs that had recently settled at a box where they had added no or only a few nest materials. A few minutes before the start of a trial, we blocked the initial nest-box occupied by the focal bird(s) and erected two boxes 4–10 m away and 2–5 m apart. Short distances were used so that the focal birds would rapidly discover the new boxes. We only did a single trial at each site and assumed that different pairs were occupying the different sites based on their spatial distribution and plumage colour.

We allowed birds to choose between the two newly erected, adjacent and identical nest-boxes with different contents (table 1). The positions of the boxes with different contents (left or right in the dyad) were chosen at random. Using pairwise tests, the effects of most confounding variables could be controlled including habitat, time of day, weather conditions, time needed by the focal birds to discover the nest-boxes, and the condition of the focal birds. To avoid human disturbance, behaviours were recorded by video filming for approximately 3 hours (range 1.1–4.9 h, table 1), starting between 6.20 and 15.20, ensuring that both boxes were within the field of view. We used digital camcorders with 32× optical zoom, on tripods placed approximately 6–10 m from the boxes. The pied flycatchers were filmed between 29 April and 31 May, the blue tits between 13 March and 21 May, and the tree swallows between 22 April and 14 May.

Each nest-box of a dyad held a dried, abandoned blue tit nest (experiment 1, on pied flycatchers), a layer of dry moss (experiments 2–8, on pied flycatchers and blue tits), or a layer of dry grass (experiments 9–10, on tree swallows; table 1). For experiment 1, the feathers added to one of the nest-boxes were taken from lining materials of other blue tit nests, using 6–10 feathers of which at least three where whitish (mostly downy breast feathers from wood pigeons) and 4–6 cm long. In the other experiments, commercially dyed chicken semi-plume feathers 4–6 cm long were used, white or black (figure 2*a,b*),

**Table 1.** Ten experiments in which three species of cavity-nesting passerine birds were presented with a dyad of nest-boxes with different contents. Reluctance to enter a focal nest-box was measured as the time elapsing from the first visit of the nest-box and the first entry of that same box (see figures 2–4 for illustrations).

| species | exp. no | nest-box content[a] | no. of trials | nest-box was never entered | duration of filming (min) mean (range) | time before entry (min)[b] median (range) | Wilcoxon test[b] Z | p | per cent of intervals >20 min[b] |
|---|---|---|---|---|---|---|---|---|---|
| pied flycatcher | 1 | blue tit nest with feathers | 37 | 9 | 167(98–224) | 11.4 (0.2–196) | | | 43 |
| | | blue tit nest without feathers | | 1 | | 1.9 (0.2–160) | −2.81 | 0.005 | 24 |
| | 2 | moss+3 white feathers | 23 | 4 | 200 (133–256) | 9.3 (0.2–120) | | | 39 |
| | | moss + 3 black feathers | | 0 | | 0.7 (0.1–177) | −2.34 | 0.019 | 17 |
| | 3 | moss + 3 white feathers | 25 | 4 | 175 (77–292) | 1.3 (0.2–291) | | | 36 |
| | | moss + 3 white paper labels | | 4 | | 1.7 (0.1–218) | −0.16 | 0.87 | 40 |
| | 4 | moss + 6 downy feathers | 25 | 3 | 178 (129–230) | 2.4 (0.1–210) | | | 40 |
| | | moss + 3 downy feathers | | 2 | | 1.3 (0.1–136) | −2.03 | 0.042 | 32 |
| blue tit | 5 | moss + 3 white feathers | 28 | 11 | 199(163–219) | 35.1 (0.6–274) | | | 61 |
| | | moss without feathers | | 1 | | 19.3 (0.1–188) | −2.32 | 0.020 | 46 |
| | 6 | moss + 3 white feathers | 27 | 7 | 189(132–224) | 56.3 (0.8–192) | | | 74 |
| | | moss + 3 black feathers | | 2 | | 29.5 (0.2–183) | −2.04 | 0.041 | 52 |
| | 7 | moss + 3 white feathers | 36 | 14 | 211(63–280) | 72.4 (1.2–240) | | | 78 |
| | | moss + 3 white paper labels | | 4 | | 30.4 (0.2–196) | −2.06 | 0.040 | 58 |
| | 8 | moss + 6 downy feathers | 26 | 6 | 215(146–274) | 53.8 (0.1–206) | | | 73 |
| | | moss + 3 downy feathers | | 5 | | 23.3 (0.1–243) | −0.60 | 0.55 | 54 |
| tree swallow | 9 | grass + 3 white feathers | 19 | 8 | 179(175–180) | 96 (5–163) | | | 74 |
| | | grass without feathers | | 2 | | 42 (0.5–144) | −2.25 | 0.024 | 58 |
| | 10 | grass + 3 white feathers | 21 | 6 | 167(165–170) | 51.1 (1.4–155) | | | 62 |
| | | grass + 3 black feathers | | 1 | | 1.1 (0.1–152) | −3.28 | 0.001 | 28 |

[a]In experiment 1, feathers from blue tit nests were used containing at least three large, whitish feathers. In experiments 2–3, 5–7 and 9–10, the feathers were commercially bought dyed chicken feathers. In experiment 4 and 8, downy feathers from wood pigeons were used.

[b]When a nest-box was never entered, the time of entry was set as when the video filming ended.

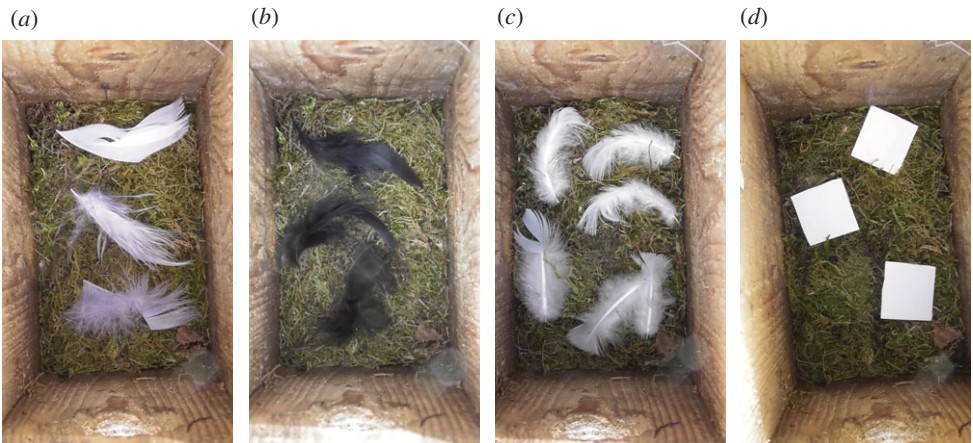

**Figure 2.** Contents of nest-boxes used in choice trials of pied flycatchers and blue tits. The nest-boxes had a base layer of dried moss and could hold either three white (*a*) or three black (*b*) commercial feathers, three or six downy feathers of wood pigeons (*c*), or three white paper labels (*d*).

except that downy feathers of wood pigeons 4–6 cm long found in the study area were used in experiments 4 and 8 (figure 2*c*). In experiments 3 and 5, we let the birds choose between a nest-box with three white commercial feathers and one with three pieces of thick, white paper ($25 \times 25$ mm$^2$), all on a layer of dry moss (figure 2*d*).

## 2.3. Feathers on the ground

Between 16 April and 15 May 2020, we offered two white and two black commercial feathers on the ground in front of 97 blue tit nest-boxes and checked the ground and the box content on the subsequent 1–4 days. The feathers were presented at the corners of a $30 \times 30$ cm$^2$ square so that all could be easily relocated. A pair of tits was only tested once either at the nest lining stage ($n = 45$) or at egg-laying ($n = 52$, 1–10 eggs laid). In 27 trials, at least one of the experimental feathers was added to the nearby blue tit nest, of which 12 (27%) trials were before, and 15 (29%) after egg-laying had started. Because of similar results, all data were combined in the statistical analyses.

## 2.4. Statistics

Different birds were studied in each experiment and so the data are independent. We defined a box visit as when a bird perched at the entrance hole, whether or not it entered the box. In the analyses, we controlled for motivation by only including trials where the entrances of both nest-boxes were visited at least once by the focal birds, and where at least one box was entered. We studied whether the focal birds avoided a nest-box of a dyad by recording the time elapsing from the first visit of the box and the first entry of the same box, and by recording whether or not a nest-box was entered at all during the trials that we accepted for analysis. When the focal bird(s) entered only one box, we assumed (conservatively) that the bird(s) would have entered the other box when the filming ended, following a previous study of ours [25]. Too few female pied flycatchers appeared, preventing analysis of female behaviour. For the blue tit and the tree swallow, where the focal birds were a pair, we did not distinguish whether the bird that visited and entered a focal nest-box was a male or female. As a measure of preference of a nest-box after inspection, we recorded the number of entries for each box after both had been entered and thus when the focal bird had gained some knowledge of the contents of each.

The time elapsing from the first visit to the first entry of the same box, could not be transformed for normality and so we used non-parametric Wilcoxon paired signed-ranks tests. Statistical tests are two-tailed with an $\alpha$-level of 0.05.

# 3. Results

## 3.1. Choice of nest-box in dyad trials

In each experiment, the birds hesitated longer to enter a nest-box with than without white feathers (figure 3 and table 1). Unmated pied flycatchers spent more time before entering a box containing a

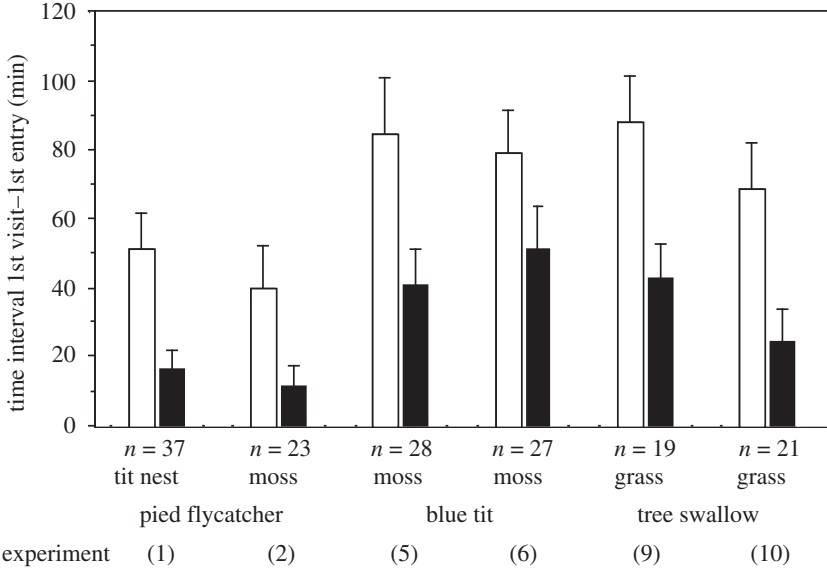

**Figure 3.** Experiments in which three cavity-nesting passerine birds were presented with a dyad of nest-boxes with different contents. Minutes (mean + s.e.) elapsing from the first visit of the focal nest-box and the first entry of that box. Values are calculated where one nest-box held three white feathers (open columns) and the alternative box had no feathers (experiments 1, 5, 9) or three black feathers (experiments 2, 6, 10). The numbers of trials are shown below the bars (*n*), and the basis layer of nest materials used in the nest-boxes. See table 1 for further explanation and statistics.

blue tit nest with than without feathers (experiment 1), and before entering a nest-box with three white, commercial feathers versus three black feathers (experiment 2). Pairs of blue tits and tree swallows spent more time before entering a box with three white, commercial feathers than one lacking feathers (experiments 5 and 9), or with three black feathers (experiments 6 and 10). For all these six experiments combined ($n = 155$), the nest-box with white feathers was never entered in 29% of the cases versus in 5% for the alternative nest-box (table 1; 45 versus 7 trials, binomial test, $p < 0.001$). In 39–74% of the experiments, the time elapsing from the first visit of the focal nest-box to the first entry of that same box exceeded 20 min for the nest-box with white feathers versus 17–58% for the alternative box (table 1).

In the pied flycatcher, the number of entries of each box after both had been entered did not differ (Wilcoxon paired sample test; experiment 1: $z = -0.30$, $n = 22$, $p = 0.77$; experiment 2: $z = -0.64$, $n = 19$, $p = 0.52$) whereas in the same test with blue tits, the box with white feathers was entered less frequently than the alternative box in one of the experiments (experiment 5: $z = -0.53$, $n = 15$, $p = 0.59$; experiment 6: $z = -2.99$, $n = 17$, $p = 0.003$). In tree swallows, the small sample size of box entries precluded analysis.

Blue tits but not pied flycatchers hesitated significantly longer to enter a nest-box with three white, commercial feathers than one with three pieces of white paper (experiments 3 and 7, figure 4 and table 1). Pied flycatchers but not blue tits hesitated longer to enter a nest-box with six versus three downy pigeon feathers (experiments 4 and 8, figure 4 and table 1).

## 3.2. Choice of offered feathers

Blue tits did not fear the commercial feathers when presented outside the nest-box. Of the feathers offered in front of 97 blue tit nest-boxes, 68% ($n = 388$) remained on the ground and 17% were taken inside the nearby nest-box (36 white and 29 black feathers, $\chi_1^2 = 1.21$, $p = 0.27$). The other 15% of feathers had disappeared. In 28% ($n = 97$) of the trials, at least one of the experimental feathers was added to the nearby nest (see illustration in figure 5 and video in electronic supplementary material).

# 4. Discussion

## 4.1. Why birds decorate their nest with feathers

We hypothesized that nest owners take advantage of the risk-averse behaviour of prospecting birds and place feathers on the surface of their nest to deceive competitors. Consistent with this idea, prospecting

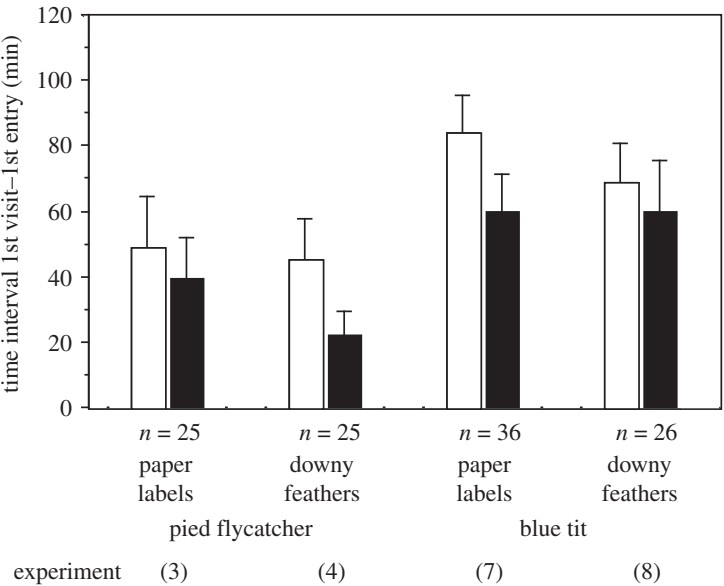

**Figure 4.** Experiments in which two cavity-nesting passerine birds were presented with a dyad of nest-boxes with different contents. Minutes (mean + s.e.) elapsing from the first visit of the focal nest-box and the first entry of that box. Values are calculated where one nest-box held three white, commercial feathers (open columns) and the alternative box had three white paper labels (black columns, experiments 3 and 7), or one box held six (open columns) and the other three (black columns) downy feathers of wood pigeons (experiments 4 and 8). The numbers of trials are shown below the bars ($n$). Moss was used as basis layer in each nest-box. See table 1 for further explanation and statistics.

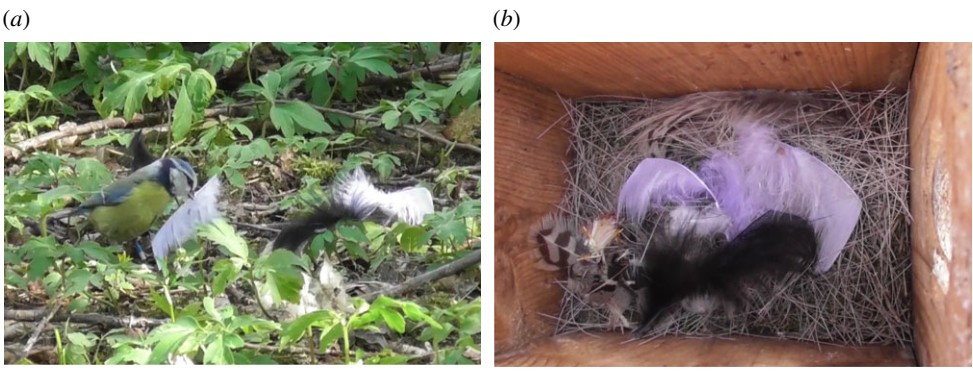

**Figure 5.** Female blue tit picking up a white commercial feather from the ground ($a$), and her nest after she had added one black and two white feathers to it ($b$). A video clip of the female when collecting the three feathers is shown in electronic supplementary material.

birds hesitated longer to first enter a nest-box with white feathers than one without feathers or with black feathers. Hesitation at the sight of white feathers was consistent across three species and families of passerine birds, that use, or never use, feathers as lining materials.

In a considerable fraction of the trials (39–74% in six experiments), it took more than 20 min for the bird to enter the box with white feathers after it first looked into that box. For these experiments, the birds did not even enter the nest-box with white feathers in as many as 29% of the trials, compared with in only 5% for the alternative nest-box. Thus, although a prospecting competitor may finally enter a nest with white feathers, the delay may allow the nest owners to return in time to avoid nest usurpation with low costs. This may be particularly important during the egg-laying period when females spend much time foraging away from the nest. In our study area, a serious competitor for tit nests is the pied flycatcher, which arrives when the tits are egg-laying and which prefers to nest on top of old or abandoned tit nests [28], even if the nest contains eggs [25]. Flycatchers claim nest sites quickly and females may start nest building within only a few hours of settlement [7] so unattended cavities are at a high risk of being claimed.

Only three conspicuous white feathers were necessary for a deterrent effect, which illustrates the potentially strong selective force of a perceived cue of predation risk. However, the fear response tended to be even stronger when six rather than three downy pigeon feathers were used in the case of the pied flycatchers. Downy pigeon feathers are light coloured and frequently used in bird nests [26]. A previous study of blue tits in our study area showed that the median number of feathers used was nine (feathers of more than 1 cm long when the nest was viewed from above [31]).

The experiment presenting white paper labels in the nest-boxes showed that the birds did not fear white objects in general but only when these were feathers; however, the result was only statistically significant in one of the two species investigated (the blue tit, figure 4). This supports the idea that the birds are reacting to a cue of predator activity in an unfamiliar cavity, but some birds may also avoid novel items, i.e. 'neophobia'. On the other hand, all three species of birds hesitated longer to enter a nest-box with white than with black feathers, probably because black feathers were less visible from the entrance of the nest-box.

Pygmy owls *Glaucidium passerinum*, which catch small birds and store the surplus in cavities, and many types of mammalian predators that use cavities, such as weasels *Mustela* spp., martens *Martes* spp. and squirrels *Tamiascurus* spp., are common in Europe and North America, so wariness of feathers in cavities may be geographically widespread. Mustelids may kill birds roosting in cavities in winter and nestlings and parents in spring, leaving feathers behind. A predator may return to a cavity where it has been successful in the past, so a prospecting bird might remain hesitant to use a cavity with feathers even after entering to check it once. However, we found that the hesitation to enter boxes with white feathers did not seem to persist long after a bird entered the box and so a strategy of using feathers to trigger a fear response may work mainly to prevent the first entry of an intruder.

The pairwise design of the dyad experiments controlled for many potentially confounding variables but we studied birds that already had settled at a nest-box and forced them to choose a new site nearby by blocking the initial nest-box. The results may therefore be conservative because the birds may already have observed that no other territory owners (or predators) were in the vicinity. We do not think the fear response is explained by our use of commercial feathers, because male pied flycatchers hesitated to the same degree with 'natural' white feathers (experiment 1, figure 3), and they also hesitated significantly longer to enter a nest-box with six versus three downy pigeon feathers (figure 4). Furthermore, the blue tits collected and used the commercial feathers we offered on the ground to decorate the nest, which shows they did not fear such feathers *per se*. However, the finding that the blue tits collected a similar number of black and white feathers from the ground indicated that feathers may be useful not only as conspicuous signals but also as nest insulation.

The Fear of Feathers Hypothesis postulates that feather decoration reduces the risk of usurping of the nest site by rivals. However, it should also be studied whether the deterrent effect of feathers in a nest is also effective against brood parasites and nest predators. Some birds add snake slough to their nest which may reduce nest predation apparently via a visual cue of a dangerous occupant [35].

## 4.2. Variation within and among species

The degree to which different species use feathers in their nests is probably innate, as shown by cross-fostering between great tits and blue tits in the wild [31]. However, in blue tits, the repeatability in the use of feathers by individual females across years is low [36] so variation may be related to feather availability and/or plastic responses to the perceived risk of nest usurpation.

The decoration of nest surfaces with conspicuous objects may risk attracting the attention of real predators that hunt by vision. Therefore, we expect the use of feathers as deceptive cues of predation will evolve when nest sites are sheltered from view, such as tree cavities, a hole on the ground, or between stones, as well as domed nests. Typically, hole-nesting birds have feathers in their nest more often than open-nesting species [37]. Small birds that nest in the open may still incorporate feathers into the structure for insulation but not for decoration. Black kites *Milvus migrants* apparently decorate their large, open nests with colourful objects to signal high quality of the nest owner to conspecific intruders [38] but such large, predatory birds may be able to defend their nest against visually hunting predators.

In the present study, male pied flycatchers entered new nest-boxes more quickly than the other two species (figures 3 and 4) perhaps because they experienced the greatest time pressure to secure a nest site quickly after arriving on the study area relatively late in spring. By contrast, the resident blue tits and migrant tree swallows may defend boxes for two to three weeks or more before starting to add nest material and probably have better knowledge of alternative nest sites. Some previous observations are

also consistent with our Fear of Feathers Hypothesis. In a study of blue tits, feathers were placed inside nearby empty boxes but no bird collected them [39]. We suggest that this was because of a fear response to a perceived risk of predation in the boxes. Our hypothesis may also help to explain why many blue tits deserted their nest when conspicuous, unfamiliar feathers were added to their nest before egg-laying [26], why decorated nests of rock sparrows *Petronia petronia* sustained fewer intrusions by floaters [19], and why spotless starlings *Sturnus unicolor* decorated their nest with more feathers at high than at low nesting density [40].

## 4.3. Alternative hypotheses for nest decoration

Collecting conspicuous objects for nest decoration may be costly and therefore an abundance of such objects may signal high quality and high resource holding potential of the nest owner [18,26,38]. However, it is not clear that feathers are rare, or costly to collect in woodland habitats [39,41]. Blue tits in our study collected only a few of the feathers that we offered on the ground close to their nest. Our dyad experiments showed that only three white feathers were sufficient to elicit a strong avoidance response in prospecting birds, which seems to be more consistent with a risk aversion than a status signalling function of feather decoration, although the hypotheses are not necessarily mutually exclusive.

It has been suggested that blue tits add feathers to the nest to signal high quality to other blue tits (an extended phenotype [26,42]) and that it is the male tit which collects the large feathers [26]. However, in our experiments, the time it took blue tits to enter boxes was not affected by the number of pigeon feathers inside (figure 4 and table 1) and, at least in Norway, Finland and Great Britain, it is only the female tit that builds, and collects material for, the nest (figure 5, [31,42,43]). Thus, it is unlikely that female blue tits add feathers primarily to signal their own quality because other blue tit pairs and prospecting male pied flycatchers, would be unlikely to take rather subtle quality differences among female blue tits into account when choosing between unfamiliar nest sites. Pairs of blue tits occupy and defend exclusive territories already in early spring and females are smaller than males and have lower social rank position [44]. The amount of decoration may signal high quality to the mate and thereby stimulate him to invest heavily in the brood. However, this hypothesis was not supported when feathers were added to nests of spotless starlings and blue tits [45,46]. Also, manipulating nest size did not affect male provisioning in blue tits [47].

Alternatively, feather decoration may simply reflect female or territory quality without having any signalling function. In Finland, the recruitment of fledgling blue tits into the local breeding population was positively correlated with a high proportion of feathers in the nest lining but not with the presence of feather 'ornaments' [21]. In this population, ornamentation of the nest with feathers seemed to be an extended phenotype of the female and not of the male [43]. In Spain, larger clutches of blue tits were in nests with more feathers, and more eggs were laid when feathers were added to the nest [26]. However, adding feathers to blue tit nests in England did not affect clutch size or breeding success [39] and in our blue tit population, there was no correlation between clutch size and the number of feathers in the nest [48].

Another possibility is that owners decorate nests with feathers simply to signal that the site is already occupied (as opposed to mimicking predator cues), and then only a few feathers may be sufficient. The addition of fresh, green vegetation to the nests of some birds of prey has been hypothesized to signal occupation [49] and also the addition of objects at the nest cavity in an owl and in a shorebird [50,51]. However, some observations do not support the 'occupation' hypothesis among the birds we studied. In a previous experiment, unmated male pied flycatchers presented with a dyad of nest-boxes more readily entered a box with exposed tit eggs than one with covered eggs [25], although the sight of clean and intact eggs should be a stronger signal of occupancy. In that experiment, we used eggs from great tits. The tit is larger than the pied flycatcher and may kill intruding flycatchers if they are caught inside its own nest [52,53]. We suggested that the bright, whitish tit eggs illuminated the otherwise dark cavity floor, making it easier for a prospecting bird to see whether a predator, or an aggressive nest owner was inside [25]. Therefore, we expected the birds in the present study to be more willing to enter a nest-box with white, reflective feathers than a dark-bottomed nest-box with no feathers, or with black feathers. However, we found the opposite, indicating that it was the visual cue of the feathers themselves that caused the fear response.

To conclude, we found that three species of cavity-nesting birds hesitated to enter strange boxes with white feathers, consistent with the hypothesis that such objects may be placed on nests by the owners to elicit a fear response both in their conspecific and heterospecific rivals and thus reduce the risk of nest

usurpation. We recommend that the Fear of Feathers Hypothesis be tested in other species as well, especially those which use concealed nest locations such as cavities, burrows, between stones or domed nests. The potential effect of decorative feathers on brood parasites and nest predators should also be investigated.

Ethics. The study complies with the current laws of Norway, and was approved by the Directorate for Nature Management in Norway (2014/2620), and by the animal welfare committee (2018/15502, 2020/23426).

Data accessibility. The data are available from the Dryad Digital Repository: https://doi.org/10.5061/dryad.41ns1rnf9.

Authors' contributions. Both authors conceived and designed the study. T.S. did the fieldwork in Norway, K.L.W. in Canada. Both analysed the respective data from the videos and both drafted the manuscript and approved the final version. Both authors carried out the fieldwork, analysed the data and wrote the manuscript.

Competing interests. We declare we have no competing interests.

Funding. Financial support was provided to K.L.W. by an NSERC discovery grant (grant no. 203177).

Acknowledgements. We thank Tarje Haug and Andreas Schillinger for assistance in the field.

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
