## [Peer Review File · Royal Society Open Science]

Review History

Decision letter (RSOS-211579.R0)

Dear Dr Slagsvold:

I am pleased to inform you that your manuscript entitled "Nest decoration: birds exploit a fear of feathers to guard their nest from usurpation" is now accepted for publication in Royal Society Open Science.

If you have not already done so, please remember to make any data sets or code libraries 'live' prior to publication, and update any links as needed when you receive a proof to check - for

instance, from a private 'for review' URL to a publicly accessible 'for publication' URL. It is good practice to also add data sets, code and other digital materials to your reference list.

on behalf of Dr Kimberley Mathot (Associate Editor) and Professor Kevin Padian (Subject Editor).

Associate Editor Comments to Author (Dr Kimberley Mathot):

Comments to the Author:

Dear Dr. Slagsvold,

I have read your manuscript carefully, and am satisfied that the experimental design, analysis and interpretation of results are all appropriate. I have a few comments/suggestions for improving the clarity of the manuscript:

Lines 61-63: Consider rephrasing to: "We propose a new idea, that birds decorate nests with feathers to trigger a fear response in their rivals, thereby reducing the risk of nest usurpation...."

Line 70: change from "often are" to "are often"

Line 75: change from "ecology and behaviour" to "ecologies and behaviours"

Line 88: begin the sentence with "To test the Fear of Feathers hypothesis, we let birds choose..."

Lines 110 to 124: I think it would be easier for readers to keep track of the two study areas if you split the descriptions into the two separate study areas/nestbox design/species, rather than try to integrate them.

Line 117 to 118: please provide the exact number of banded blue tits out of the total sample of blue tits.

Line 129-130: Were all singing males assumed to be unmated? Please clarify how you determined individuals to be "unmated".

Line 143: delete "of the day"

Line 161: replace "During" with "Between"

Line 181: add "preventing analysis of female behaviour".

Line 183: replace "popularity" with "preference".

Thank you for transferring your article for consideration for publication at Royal Society Open Science.
